# Definition of Optimized Indicators from Sensors Data for Damage Detection of Instrumented Roadways

**DOI:** 10.3390/s22155572

**Published:** 2022-07-26

**Authors:** David Souriou, Jean-Michel Simonin, Franziska Schmidt

**Affiliations:** 1Laboratory for Modelling, Experimentation and Survey of Transport Infrastructures (LAMES), Department of Materials and Structures (MAST), Univ. Gustave Eiffel, F-44344 Bouguenais, France; 2Urban and Civil Engineering Testing and Modeling Laboratory (EMGCU), Department of Materials and Structures (MAST), Univ. Gustave Eiffel, F-77447 Marne-la-Vallée, France; franziska.schmidt@univ-eiffel.fr

**Keywords:** pavement instrumentation, signal processing, indicators, weighting functions

## Abstract

Pavement instrumentation sensors have become a common practice for structural monitoring. Data processing often consists of extracting different values generally representative of variations in the whole pavement, but this makes it more difficult to follow a specific characteristic of the structure. To overcome this limitation, this paper aimed to transpose an original method, labeled the Optimized Indicators Method (OIM), which consists of finding some weighting functions of a signal to evaluate some indicators linked to a physical characteristic of the structure. The main advantage of this method is that the inferred indicators are particularly sensitive to a specific characteristic of the structure without being sensitive to the others. This research work consisted of analyzing, conventionally and with the OIM, the signals of strain gauges, which were recorded during an experimental campaign carried out on a circular instrumented pavement and submitted to an accelerated fatigue testing. The OIM, calibrated through an experimental reference signal, makes it possible to evaluate independently, through weighting functions, some specific physical characteristics of the pavement. It showed a two-stepped degradation, starting with a damaging of the bituminous layer, followed by an alteration of the base layer, which could not be easily deduced from a conventional processing method.

## 1. Introduction

Roadways correspond to the most strategical infrastructures for the transport and communication, and their structural engineering is country-specific to answer the needs of the increase of heavy traffic [1]. Pavements are structures with various layers, generally a wearing course layer above the base and subbase layers, which have specific characteristics. The mechanical and environmental actions lead to the creation of defects such as cracks, potholes, and rutting, which decrease the service life of these infrastructures and affect the safety of users. When these defects appear on the surface, a large part of the bituminous layer is generally destroyed and replaced by a new one, which generates unacceptable costs. Hence, the use and the development of Non-Destructive (ND) methods and the pavement’s asset management systems are of strong interest [2,3]. Ground Penetrating Radars (GPR) and Falling Weight Deflectometers (FWD) are currently among the most used external evaluation techniques for a global characterization of the structure [4,5]. GPRs allow proceeding to a global auscultation of the pavement through electromagnetic waves to evaluate the thickness of the different layers or to detect (and eventually to classify) cracks if the frequency is high enough [6,7]. FWDs are another current ND method, generally used to evaluate the bearing capacity of roads, and consist of the measurement and the processing of deflection basins, which allow evaluating structural condition rating [8] or mechanical properties of pavements [9]. However, GPR and FWD generally suffer from the need for considerable equipment, lack of precision for the detection of microdefects, and they can be sensitive to the environment. Pavement instrumentations including optic fiber sensors, pressure sensors, strain gauges, accelerometers, wireless embedded sensors, etc., make it possible to continuously and passively collect data for the monitoring of structures [10]. Moreover, like GPR and FWD, these methods require the management of huge databases, and their design must be adapted to the structure. The data processing of the aforementioned methods often consists of extracting different values, such as extremum, number of positive or negative peaks, and in calculating thresholds, that are generally representative of variations in the whole structure. Therefore, it can be difficult to detect a defect located in the base layer and to prevent its propagation through the bituminous layer, or vice versa. An early detection of a degradation occurring in a specific layer of a pavement would allow preventing further damage of the structure, pointing to the need for a simple and reliable method to evaluate separately and independently the characteristics of each layer.

To overcome this limitation, Le Boursicaud [11] developed an original method that consists of finding a Weighting Function of the signal that is particularly sensitive to a specific characteristic of the structure without being sensitive to the others.

This method, referenced as Optimized Indicators Method (OIM), was successfully applied for the processing of FWD signals by Simonin et al. [12], and showed that the OIM makes it possible to evaluate separately the physical characteristics of the different layers of a pavement. In this case, the construction of the weighting functions with the OIM consists of identifying, in the first step, an analytical model that simulates the ND method’s signal by considering the physical characteristics of the studied structure. A reference signal (obtained experimentally on a real pavement on its initial state) is chosen and used to calibrate the model, which allows for evaluating the reference values of the physical characteristics of each layer of the pavement. The model then makes it possible to separately simulate the signals that would be obtained after a slight variation of a specific pavement’s characteristic. These simulated signals are then proceeded through integrals and derivatives in a second step to deduce the weighting functions. Once applied to real FWD signals, the OIM allows independent evaluation of the pavement characteristics as long as their variations remain in the linearity domain.

As the OIM consists of evaluating weighting functions through simulated signals representing the response of a pavement subjected to a mechanical solicitation, one can ask if this method could be used for the data processing of other ND methods, such as embedded sensors. Recently, data were recorded from an instrumented pavement during an experimental campaign simulating high traffic running loads through an accelerated pavement testing facility (fatigue carousel) located at Univ. Gustave Eiffel, campus of Nantes (Univ. Eiffel Nantes, previously known as IFSTTAR Nantes) [13,14]. The tested pavement was instrumented with strain sensors and temperature sensors to evaluate, among other things, the fatigue resistance of different formulations of asphalt materials. Thus, with these available data, we seized the opportunity to check a possible implementation the OIM for the data processing of strain gauges embedded in an instrumented pavement. The aim of this paper consisted, then, of highlighting the possibility to extract some weighting functions from embedded strain sensors signals for the separate monitoring of the physical characteristics of the studied pavement. The second objective was showing that the OIM can bring additional information compared with more conventional strain sensors signal processing.

In the Section 2, the principle of the OIM and the weighting functions is briefly explained. Section 3 describes the instrumented pavement, the fatigue carousel, and the experimental setup. In Section 4, numerical modeling of the pavement behavior under the experimented conditions is proposed, which allows for the calculation and the validation of the weighting functions in Section 5. To complete this study, the data recorded from the instrumented pavement were analyzed by both extracting classic indicators and processing with the OIM, and the results were then compared and discussed.

## 2. Theoretical Background of the Optimized Indicators Method (OIM)

This method, developed and applied by Le Boursicaud [11] to obtain optimized indicators from deflection basin measurements, consists of searching for weighting functions from simulated signals. As constructed, these weighting functions are sensitive to the variations of a specific characteristic of the pavement structure (elastic modulus or thicknesses of the layers, interface status, etc.), as long as they remain in a linearity domain. So, in order to monitor the evolution of these characteristics during the pavement service life over time, a synthetic indicator that represents one of the pavement’s characteristics is calculated from a complete signal. This section proposes a concise presentation of the principle of this method, based on the calculation of derivatives and integrals; more details can be found in the study published by Simonin et al. [12]. In this paper, the mathematical expressions were written with the hypothesis of summation over the repeated indices. Therefore, they involve general vectors of coefficients.

From a simulated signal (gauge signal, deflection bowl), $\varepsilon \left(x,Y_{n}\right)$, which depends on an acquisition parameter *x* (time, distance…) and n pavement characteristics $Y_{n}$ (*n* = 1, …, *j*, *k*, …), the problem consists of looking for a set of indicators $I_{n}$. The determination of an indicator $I_{j}$ takes the form of an integral characterized by the function $p_{j}\left(x\right)$, designed as the weighting function, which is essentially sensitive to the value of the pavement characteristic $Y_{j}$, as indicated by the following equation:(1)$$I_{j}=\int_{-x}^{+x}p_{j}\left(x\right)\varepsilon \left(x,Y_{n}\right)dx$$

The determination of the weighting function $p_{j}\left(x\right)$ was conducted at the vicinity of a set of reference parameters $\left(\ldots \;Y_{j},\;Y_{k}\ldots \right)$. Hence, the sensitivity of $I_{j}$ to the pavement characteristic $Y_{j}$ can be assessed from a partial derivative calculated as:(2)$$\delta I_{j}=\int_{-x}^{+x}p_{j}\left(x\right)\frac{\partial \varepsilon \left(x,Y_{n}\right)}{\partial Y_{j}}\;\delta Y_{j}dx$$

The weighting function, $p_{j}\left(x\right)$, is in the function vector space generated by the n functions and can be written as a linear form with a real coefficient matrix, $a_{jk}$, as follows:(3)$$p_{j}\left(x\right)=a_{jk}\frac{\partial \varepsilon \left(x,Y_{n}\right)}{\partial Y_{k}}$$

So:(4)$$\begin{array}{ll} \delta I_{j}= & \int_{-x}^{+x}a_{jk}\frac{\partial \varepsilon \left(x,Y_{n}\right)}{\partial Y_{k}}\frac{\partial \varepsilon \left(x,Y_{n}\right)}{\partial Y_{j}}\;\delta Y_{j}dx\\ & =\left(\int_{-x}^{+x}\frac{\partial \varepsilon \left(x,Y_{n}\right)}{\partial Y_{j}}\frac{\partial \varepsilon \left(x,Y_{n}\right)}{\partial Y_{k}}\;dx\right)\;a_{jk}\;\delta Y_{j} \end{array}$$

Which finally leads to:(5)$$\delta I_{j}=b_{jk}a_{jk}\;\delta Y_{j}$$

With:(6)$$b_{jk=}\int_{-x}^{+x}\frac{\partial \varepsilon \left(x,Y_{n}\right)}{\partial Y_{j}}\;\frac{\partial \varepsilon \left(x,Y_{n}\right)}{\partial Y_{k}}dx$$

The orthogonality of $p_{j}\left(x\right)$ is imposed with respect to the functions $\frac{\partial \varepsilon \left(x,Y_{n}\right)}{\partial y_{j}}$, with j≠k as:(7)$$a_{jk}b_{jk}=1\;\mathrm{for}\;j=k$$
(8)$$a_{jk}b_{jk}=0\;\mathrm{for}\;j\neq k$$

The determination of the optimized indicator can be achieved following the procedure described in the flowchart (Figure 1).

As described, the derivatives and integrals of the simulated signal $\varepsilon \left(x,Y_{n}\right)$ were calculated numerically so that the coefficients of the matrix $b_{jk}$ were obtained first (with Equation (6)). Through the orthogonality condition given by Equations (7) and (8), it is possible to deduce the coefficients of the matrix $a_{jk}$, which allows determining $p_{j}\left(x\right)$ (Equation (3)) and, finally, the indicator $I_{j}$. After the calculation and the validation of the weighting function, the optimized indicator $I_{j}$ was normalized through a reference indicator *I_R_* calculated from a reference signal $\varepsilon_{R}\left(x\right)$. This reference signal was obtained through numerical modeling (or a real measurement) of the considered structure.

The Section 3 describes the structure of the instrumented pavement, the fatigue carousel, and the experimental procedure that was used for the acquisition of data for implementation of the OIM for embedded strain gauges.

## 3. Description of the Experiment

### 3.1. The Pavement Fatigue Carousel

The fatigue carousel (Figure 2) at Univ. Eiffel Nantes is an outdoor road traffic simulator designed to study the behavior of real-scale pavements under accelerated heavy traffic [15,16]. The fatigue carousel has a diameter of 40 m and four loading arms, which can each carry dual-wheel loads able to apply a charge up to thirteen tons, during spinning cycles at speeds up to 100 km/h. Two months of testing can represent up to 20 years of heavy traffic undergone by a moderate traffic pavement (150 heavy trucks/day). During the spinning cycles, a lateral wandering of the loads can be applied to simulate the lateral distribution of loads of real traffic [17].

### 3.2. Tested Instrumented Structure

The pavement considered in this paper is the fifth section from the previous study from J. Blanc et al. [18]. It consists of two layers, namely a surface bituminous layer and a base layer. The bituminous layer is 10.2 cm thick and made of a bituminous material with a high modulus asphalt mixture (HMAM, aka. EME2 in the previous study [18]) with 20% of reclaimed materials. The base layer is 76 cm thick, made of Unbound Granular Material (UGM) on a stone bed (50/120 mm). Figure 3a represents the schematic of the pavement structure. The pavement was, among others, instrumented with longitudinal strain gauges Dynatest PAST-IIA (Figure 3b) implemented at the bottom of the bituminous layer, at 9 cm depth (so, 1 cm above the Unbound Granular Material (UGM)) during the construction.

The strain gauges were designed for measurement in asphalt concrete and consist of a bar of fiberglass reinforced with epoxy, in which is inserted a gauge of deformations. This bar was then protected with different materials and fixed to a H-shaped anchoring system. The measurement range of this type of gauge is ±1500 µ strain (micro strain, corresponding to the strain value multiplied by 10^−6^) with a sensitivity of 0.11 N/µstrain and capable of withstanding the high temperature of the hot asphalt concrete during construction (up to 150 °C) [19]. In addition, some temperature sensors (not represented in Figure 3 for readability purposes) were also embedded at 0 and 10 cm depth in the bituminous layer so that this parameter could be monitored during this experimental campaign. All signals were recorded as a function of time at a frequency of 600 Hz.

### 3.3. Loading Characteristics and Data Acquisition

After the construction of the pavement, a first series of spinning charging cycles was applied as a consolidation step of the pavement. The amount of cycles applied during this consolidation step was 37,800, which, considering the four arms of the carousel, corresponds to a total amount of N = 4 × 37,800 = 151,200 loadings. The load applied by the dual-wheel load during the entire experimental test was 65 kN, which corresponds to the maximum load allowed on pavements in France. Loading wheels can be located at 11 positions along the transverse profiles of the pavement to simulate the lateral wandering of the traffic.

In this study, only measurements recorded with dual loads located at position 6, which corresponds to the center of the wheel path, were considered. In this configuration (Figure 3a), the center of the two twin loading wheels passed above the embedded strain gauge during the spinning cycles. The signal of the gauge (longitudinal strain, given in µ strain, corresponding to the strain value multiplied by 10^−6^) consists of a measurement as a function of time of the strain that reached its maximal value (peak, indicating an extension of the gauge) each time that a dual-wheel load passed above the sensor, as pictured in Figure 4.

This signal being recorded at a frequency of 600 Hz, and the velocity of the loading arms being known, makes it possible to deduce how far the dual-wheel load is from the gauge considering the time associated with the peak as the zero distance. Hence, we chose to represent the signal of the gauge as a function of the position of the loading arms for distance ranges arbitrarily set between −3 and +3 m. Then, the measurement of the longitudinal µ strains at an average temperature of 9 °C and a loading speed of 72 km/h after 151,200 loadings was recorded as a reference signal. The recorded reference signal was assumed as an initial state without damage to the structure. Indeed, at N = 0, some phenomenon of post-compaction of the layers may occur afterwards, which could influence the evolution of the modules of the layers.

In the next step, an equivalent of a total traffic corresponding to 881,600 heavy loads was applied over four months (from mid-November 2017 to the beginning of February 2018). The recorded strain gauge signals correspond to the strain measured from −3 to +3 m of distance of the gauge along the longitudinal direction of dual wheel loads. Through a regular recording of the gauge’s signal, it is possible to study the evolution of the strain gauge signal (and of some specific indicators) as a function of the number of applied loads. During this entire fatigue test, the pavement surface temperature varied between −6 °C and 27 °C, with most values between 7 and 12 °C. The mean surface temperature was 9.3 °C, and the mean temperature in the middle of the bituminous layer was 9.4 °C [18]. Figure 5 represents the mean temperatures and their respective dates of acquisition as a function of the number of applied loads for informative purposes (the strain gauge signals being shown and discussed further in this paper).

The aim of this experiment was to process the strain gauge data through the previously described OIM in order to deduce the evolution of specific indicators as a function of the number of applied loads. The construction of the weighting functions requires the use of a model simulating the behavior of the structure considering the physical characteristics of each layer and depending on the loadings (including temperature). The reference signal obtained at N = 151,200 loadings serves as a base for the numerical calibration of this model. The Section 4 describes the model and the numerical tool used to simulate the strain gauge signal.

## 4. Modelling

### 4.1. Huet-Sayegh Model

The Huet–Sayegh model is recognized to fit the linear and viscoelastic behavior of bituminous materials well [16,20]. This model is represented in Figure 6 by a purely elastic spring ($E_{0}$) (Branch I) connected in parallel to two parabolic dampers in series with an elastic spring ($E_{\infty }-E_{0})$ (Branch II) [21,22].

In the frequency domain in which asphalt materials are characterized, the complex modulus of the Huet–Sayegh model reads:(9)$$E^{*}\left(\omega ,\tau \left(\theta \right)\right)=E_{0}+\frac{E_{\infty }-E_{0}}{1+\delta {\left(i\omega \tau \left(\theta \right)\right)}^{-k}+{\left(i\omega \tau \left(\theta \right)\right)}^{-h}}$$
where $E_{0}$ is the static elastic modulus, $E_{\infty }$ is the instantaneous elastic modulus, *k* and *h* are exponents of the parabolic dampers (1 > *h* > *k* > 0), and *δ* is a positive non-dimensional coefficient balancing the contribution of the first damper in the global behavior. The parameter *θ* denotes temperature and *τ* is a response time parameter [23], which accounts for the equivalence principle between frequency and temperature of bituminous materials, governed by:(10)$$\tau \left(\theta \right)=\exp\left(A_{0}+A_{1}\theta +A_{2}\theta^{2}\right)$$
where $A_{0}$, $A_{1}$ and $A_{2}$ are constant parameters.

To simulate a reference gauge signal, all parameters are calculated numerically.

### 4.2. VISCOROUTE© Software

VISCOROUTE© is an analytical, multilayer pavement modeling software, based on the viscoelastic model of Huet–Sayegh, developed at Univ. Eiffel Nantes for the modeling of pavements with bituminous materials under moving wheel loads [24]. The pavement described in Section 3.2 is modeled as a multi-layer structure with bonded interfaces, lying on bedrock with infinite stiffness. Thus, the model also takes into account the elastic modulus of the UGM layer ($E_{UGM}$). The simulations were carried out under the conditions of temperature and speed of the experimental campaign and the model adopted a viscoelastic behavior of bituminous material integrated into the VISCOROUTE© 2.0 software [24,25,26].

After analyzing and comparing the influence of all parameters on the shape of the signal, and based on the results from mechanical characterizations of the bituminous and base layers (not shown in this paper), the Huet–Sayegh model was numerically adjusted so that the simulated signal fit the measured reference signal. The signal corresponds to the strain measured from −3 to +3 m of distance of the gauge along the direction of dual wheel loads. The Table 1 presents the optimized pavement parameters obtained from the numerical model.

Figure 7 shows the shape of both reference and numerically modeled signals (µ strain as a function of the distance at which the dual-wheel load is from the sensor as discussed in Section 3.3).

As suggested by the labels in Figure 7, it is possible to extract from these signals the maximal and minimal values or the distance between the two minimal values (width). The role of these values is discussed further.

Concerning the reference gauge signal, the longitudinal strain is in contraction just before the wheel passes (indicated by the negative value of the strains), in extension under the passing wheel (signified by positive strain), and then in contraction again. The experimental reference signal is not symmetrical; we assume that this dissymmetry is due to the viscoelasticity of the bituminous materials. The dissymmetry is nearly respected with the viscoelastic model, as the two minimal values were respectively −25.81 and −26.42 µ strain. The proposed numerical model was kept and serves as a basis for calculating weighting functions and optimized indicators in Section 5.

## 5. Calculation of the Optimized Indicators and the Weighting Functions

### 5.1. Choice of the Construction Parameters of the Optimized Indicators

For the construction of the optimized indicators, a parametric study was carried out using the VISCOROUTE^©^ software to check the influence of each pavement parameters on the shape of the modeled signal. The parametric study consists of separately applying a ±10% variation on each parameter of the model (indicated in Table 1) by considering the other parameters as constants. With respect to the previously proposed model and considering the values of parameters listed in Table 1 as references, each of these variations differently affects the shape of the signal and the value of the equivalent module (|*E**|). The aim of this parametric study was identifying the two most influential parameters usable to determine separately the characteristics and/or the damage level for two different layers. The determination of these influential parameters consists of a first step in a numerical integration of the simulated reference signal to calculate its surface ($S_{ref}$). In a second step, after applying the ±10% variation on a chosen parameter, a second surface ($S_{param}$) was deduced by integration of the simulated signal. Then, the difference between these two surfaces over $S_{ref}$ allowed the calculation of a relative surface variation %S as given by the following equation:(11)$$\%S=100\frac{S_{param}-S_{ref}}{S_{ref}}$$

Table 2 regroups the values of relative surface variation obtained after applying a ±10% variation for each parameter of the Huet–Sayegh model.

It is theoretically possible to calculate the weighting functions for all the parameters, but the results of the parametric study showed that $E_{\infty }$ and $E_{UGM}$ influence the shape of the signal the most. Figure 8 shows the influence of these two parameters on the shape of simulated signal for illustrative purposes.

As shown in Figure 8a, with the increase/decrease of $E_{\infty }$, the maximum and minimum indicators relatively decreased/increased while the width remained nearly unchanged. In the case of an increase/decrease for $E_{UGM}$ (Figure 8b), the most obvious changes consisted of a decrease/increase of the maximum indicator (in a lesser proportion compared with those observed for $E_{\infty }$) and a slight tightening/spreading of the width indicator, with no significant changes being observed for the minimum.

The other parameters of the Huet–Sayegh model having a lesser effect on the shape of the simulated signal (see Table 2), $E_{\infty }$ and $E_{UGM}$, were selected for the determination of the optimized indicator and the calculation of the weighting functions.

### 5.2. Choice of the Construction Parameters of the Optimized Indicators

The weighting functions for $E_{\infty }$ and $E_{UGM}$ were calculated in accordance with the method described in Section 2. The weighting functions were determined from the reference signal of the chosen model calibrated with VISCOROUTE^©^ and by separately making a ±10% variation of the two selected parameters. Using Equation (6), a 2 × 2 $b_{jk}$ matrix was constructed from the simulated data, and the orthogonality between $b_{jk}$ and $a_{jk}$ allowed for calculating the latter (Equations (7) and (8)); then, the weighting function associated with the selected parameters and for each position of *x* was deduced (Equation (3)). Figure 9 shows the weighting functions calculated from the reference signal for the two selected parameters, labeled as $P_{E_{\infty }}$ and $P_{E_{UGM}}$.

The interpretation of the weighting functions is not intuitive but the two curves had a nearly opposed shape (meaning they could partially offset one another depending on the distance *x*), and different scale based on the reference value of $E_{\infty }$ and $E_{UGM}$. The validation of these weighting functions requires a sensitivity study based on the calculation of the optimized indicator deduced from simulated signals with ±10% variations of the considered parameters. We indeed assume that the pavement characteristics are in the linearity domain as long as their variation remains below the ±10% variation range.

The integrals of the product between the weighting functions and the reference or the simulated signals with ±10% variation of $E_{\infty }$ (and $E_{UGM}$) were calculated according to Equation (1). This allows deducing the reference and simulated optimized indicators labeled as $I_{E_{\infty }}$ (and $I_{E_{UGM}}$). To obtain the relative deviation, the difference between the reference and the simulated indicators was divided by the value of the reference-optimized indicator. Considering the reference signal and the signals simulating ±10% variations of the considered parameters, Table 3 lists the calculated indicators and their relative deviations.

The integral between the reference signal and the weighting functions associated with $E_{\infty }$ and $E_{UGM}$ allows for deducing the reference indicators, $I_{E_{\infty }}$ and $I_{E_{UGM}}$ (Equation (1)). The first result shown in Table 3 is that a variation of 10% of the module ($E_{\infty }$ and $E_{UGM}$) led to a deviation in the range of ±10% on the optimized indicators (the deviation being relatively greater in the case of $E_{\infty }$). The second result validates the orthogonality of the method as a ±10% variation for $E_{\infty }$ ($E_{UGM}$), leading to variation in the range of ±0.25% for $I_{E_{UGM}}$ (0.13% for $I_{E_{\infty }}$). This highlights a satisfying sensitivity for the weighting functions and the calibration of the OIM as both parameters can be evaluated independently each other and with accurate variation ranges.

The final step, discussed in the Section 6, consists of applying the calibrated OIM on the experimentally measured signals. The aim was comparing the indicators conventionally extracted from the experimental signals to those deduced from the OIM.

## 6. Data Processing of the Carousel Test

### 6.1. Evolution of Signals’ Values with Loadings

Figure 10 shows the evolution of the experimental gauge signals shape as a function of the number of loadings. For reminding purposes, the beginning of this study corresponds to the signal recorded at N = 151,200 loadings (reference signal) and ends at N = 881,600 loadings.

Figure 10 shows that the amplitude of the signals, and thus the strain measured by the sensor, increased with the number of loads, which indicates a progressively increasing fatigue of the structure. The maximum value measured by the sensor at each loading step is a conventional indicator used to monitor the bearing capacity of the pavement with the load [27,28]. As suggested by the labels on Figure 7, it is possible to extract three different indicators from the measured signals identified as the maximum, the minimum (two values), and the distance (width) between these two minimum values. With the increase of the number of loads, the maximum value (and the absolute value of the two minimum peaks) of the signal increased, while the width tended to tighten. These values, designed as indicators, were normalized with respect to those extracted from the reference signal (N = 151,200 loadings).

Figure 11 shows the evolution of the three normalized aforementioned indicators (in %) with the number of loads. The two lines indicating a relative variation of ±10% allow for identifying at which number of loads the indicators may start to be out of the linearity domain.

A closer comparison between the maximum and the width indicators curves allowed for observing a similarity of their both shapes, one being the opposite of the other (similar increasing and decreasing tendencies and steps) and with different scales. Thus, the increasing tendencies for the maximum indicator (and eventually the decreasing tendency for the width indicator) could be translated in terms of decreasing of the bearing capacity of the structure, but without a precise diagnosis of the damaged level of each layer.

The shape of the minimum indicator started to differ from the maximum indicator at N = 652,000, which could indicate another unidentified phenomenon occurring in the pavement structure. The step observed around N = 789,200 for each indicator could characterize a damaging in the whole structure.

To obtain a more detailed knowledge on the concordance between the variations of these indicators and phenomena that may occur in the pavement structure (damage, viscoelasticity…), a characterization of the structure at each loading cycle would be necessary to follow the variation of the parameters of the model of Huet–Sayegh. This justifies the use of the OIM for the processing of the strain gauge signals to attempt evaluation of the characteristics of each layer separately.

### 6.2. Implementation of the OIM

The implementation of the OIM on all recorded signals (see Figure 10) allowed for extracting a value of the optimized indicator of the considered pavement parameters (respectively labeled as $I_{E_{\infty }}$ and $I_{E_{UGM}}$) for each number of applied loads. The ratio between the reference (N = 151,200) and the extracted optimized indicators allowed for representing the evolution of normalized pavement parameters with respect to the number of loads, as plotted in Figure 12. Similar to Figure 11, relative variations in the range of ±10% are indicated by two lines.

The optimized indicator $I_{E_{\infty }}$ follows a decreasing tendency with a shape similar to the width (or maximum in line with the discussion of the previous paragraph) indicator as plotted in Figure 11. Its relative variation was greater than 10% at N = 245,600, similarly to the maximum and minimum indicators plotted in Figure 7. At the beginning of the tests, and according to simulations realized on the reference signal (see Table 1), $E_{\infty }$ was estimated at 19,000 MPa, which corresponds to 100% for the optimized indicator in a normalized scale. According to the OIM, this value decreased to 17,100 MPa (90%) after 207,600 loadings, which indicates that the bituminous layer started to become damaged with repeated loadings, decreasing the bearing capacity of the structure, which is physically consistent. However, as $I_{E_{\infty }}$ had a decrease in the range of 20% at 245,600 loadings, one can assume that the obtained results are out of the linearity domain; therefore, results obtained above N = 245,600 may be debatable. Nevertheless, at 881,600 loadings, $I_{E_{\infty }}$ was evaluated at 33.95%, which would correspond to nearly 6791 MPa for $E_{\infty }$ (based on the initial value of $E_{\infty }$ evaluated at 19,000 MPa)

The optimized indicator $I_{E_{UGM}}$ remained constant and in the range of the +10% relative variation, from the beginning to N = 652,000, indicating that the base layer was not affected by repeated loadings. However, a net increase of 50% occurred in the range of 688,400 loadings. With the degradation of the bituminous layer, the repeated loadings can start affecting the base layer and result in an increase of its elastic modulus. As $I_{E_{\infty }}$ is out of the linearity domain at 652,000 loadings, the evaluation of the variation of $I_{E_{UGM}}$ is certainly erroneous as the weighting function $P_{E_{UGM}}$ was calibrated with $E_{\infty }$ having a constant value of 19,000 MPa. This can explain the exaggerated variation of $I_{E_{UGM}}$ up to 160% for N = 881,600, corresponding to $E_{UGM}$ = 191 MPa (based on the initial value evaluated at 120 MPa through VISCOROUTE©). The results, in any case, demonstrate the interest of the OIM as each considered parameter of the pavement could be evaluated separately at least until N = 652,000.

Thus, at this stage of this research, one can assume that once the variation of a pavement characteristic reaches a given threshold, a recalibration of the weighting functions based on a new reference signal would restore the linearity of the method. In the next Section, we propose to check the validity of this hypothesis.

### 6.3. Data Processing through Regular Recalibration of the OIM

The principle of this alternative data processing consists of re-calculating the weighting functions from the measured signals each time that one of the optimized indicators had a relative variation greater than 10%. As observed in Figure 12, $I_{E_{\infty }}$ had a variation greater than 10% at 245,600 loadings, which makes it possible to assume that the weighting function is out of the linearity domain. Before any new calculations, the values of $E_{\infty }$ and $E_{UGM}$ were calculated and saved from 151,200 to 207,600 loadings. Indeed, in this loadings range, the weighting functions remain in the linearity domain of the first calibration. Afterwards, the gauge signal recorded at 245,600 loadings (labeled as N_ref_ = 245,600) was selected and used as a new reference signal for the recalibration of the weighting functions. All steps described in Section 4 and Section 5 were repeated to calibrate new weighting functions that were then applied on all remaining signals recorded from 245,600 to 881,600 loadings, allowing the plotting of new values for the optimized indicators as a function of the number of loads. With these recalibrated weighting functions, another deviation greater than 10% was observed for $I_{E_{\infty }}$ at 450,400 loadings. The evaluated values of $E_{\infty }$ and $E_{UGM}$ obtained from 245,400 to 429,600 loadings were saved, and a third calibration was conducted with the signal recorded at N = 450,400 loadings as the new reference. This procedure was repeated each time one of the deduced optimized indicators had a relative variation greater than ±10%. In this study, three recalibration procedures were required to proceed all gauge signals with the OIM, and were performed at 245,600, 450,400, and 652,000 loadings.

Figure 13a,b represent respectively the values of $E_{\infty }$ and $E_{UGM}$ deduced from this recalibration procedure, with their respective ±10% variation windows for illustration purposes, and compared with the values obtained without regular recalibration.

The two elastic moduli evaluated with regular recalibration followed a tendency slightly similar from those without recalibration until 688,400 loadings. This result suggests that the chosen model remains in the linearity domain for relative variations of the parameters greater than ±10%.

At N = 881,600, the final evaluated value for $E_{\infty }$ with regular recalibration was 6045 MPa, which is lower in comparison of the 6790 MPa evaluated without regular recalibration. This would point a slightly more severe damaging of the bituminous layer with repeated 65 kN loadings. The $E_{UGM}$ evaluated with regular recalibration constantly remained in the ±10% variation windows, with a net increase from N = 688,400 to 789,200. The final evaluated value was 137 MPa, which seems more realistic than the 191 MPa evaluated without regular recalibration. These results are more deeply discussed in the next Section.

### 6.4. Discussion and Prospects

An interpretation of the results shown in Figure 13 could be as follows:-The bituminous layer is progressively damaged during the complete experimental campaign, this damage leading to a decrease of its instantaneous elastic modulus. After a regular decrease from 152,200 to 536,000 loadings, this characteristic remained nearly constant in the range of 652,000 to 789,200 loadings and noticeably decreased at 820,000 loadings.-The base layer, composed of unbound granular materials, kept a nearly constant 120 MPa elastic modulus from 151,200 to 688,400 loadings, before a net increase evaluated up to 130 MPa at N = 789,200. As the elastic modulus of the bituminous layer is reduced due to the repeated 65 kN-loadings, we assume that a slight compaction of the base layer may occur through compressive stress transmission. Afterwards, the damaging of the bituminous layer continued at 820,000 loadings, while the base layer was unaffected.

Considering Figure 11, the maximum indicator remained constant from 688,400 to 789,200 loadings while the two other indicators continued their decreasing/increasing tendencies, pointing to a critical event in the structure, difficult to interpret in a conventional way. Through the results extracted from the OIM, this phenomenon seems to indicate an alteration of the base layer at 688,400 loadings caused by the progressive damaging of the bituminous layer.

Some pictures, that show some transversal cracks merging on the surface of the bituminous layer after a given number of loads, were taken at the end of the experimental campaign during the previous study [14,18]. These pictures confirm the progressive degradation of the bituminous layer. Unfortunately, no pictures of the UGM layer after this experimental campaign are available for this study.

The results of this study showed, in any case, that the OIM could be used for the processing of strain gauge data to separately monitor the characteristics of each layer of a pavement. The interest of the OIM demonstrated as the conventional processing of signals showed that a single proposed indicator did not make it possible to follow explicitly the evolution of a particular parameter of the pavement.

As a prospect, the evaluation of other parameters with the OIM would allow establishing a correlation between each of them and eventually deduce other phenomena such as an alteration of the viscoelasticity.

The determination of a threshold variation appears as a key parameter for the re-calibration of the weighting functions. A clear identification of the linearity domain of the OIM would require carrying out another parametric study consisting of evaluating the influence of the model parameters with greater relative variations (±30%, for example).

A successful implementation of the OIM on the data processing of strain gauge signals would open the opportunity to transpose the method for the treatment of other signals such as those from geophones or generated from other Non-Destructive methods used for the characterization of different civil engineering structures.

## 7. Conclusions

Through the recent generalization of pavement instrumentation as a structural monitoring technique, the conventional data processing methods generally only allow for evaluating the global characteristics of these structures. In this paper, we successfully transposed a data-treatment method, labeled as the Optimized Indicator Method (OIM). This method consists of extracting some weighting functions, through derivatives and integrals, from a modeling of the structure to evaluate its physical characteristics independently from each other.

The considered structure was a pavement, made of a bituminous layer and an Unbound Granular Material layer, instrumented with embedded longitudinal strain gauges. The first step of the study consisted of recording the gauges’ signals as a function of 65 kN loadings applied through a fatigue carousel simulating high heavy traffic. After the choice of a reference signal, the VISCOROUTE© software allowed, for numerically setting the parameters of the model of Huet–Sayegh, identifying the most influential parameters on the shape of the signal. Then, after deduction of the associated weighting functions, proceeding with the OIM of signals recorded during nearly 4 months at regular steps showed that the elastic modulus of the bituminous layer regularly dropped while the elastic modulus of the base layer remained nearly constant. After reaching a critical number of loads (688,400 loadings), the base layer seemed affected as its elastic modulus increased from 120 to 130 MPa, and the degradation of the bituminous layer continued.

These results demonstrate the interest of the OIM, as the conventional methods cannot easily and separately evaluate the physical characteristics of the pavement layers. However, a regular recalibration of the method seems required as long as at least one of the pavement characteristics reaches a relative variation out of the linear domain. This first result opens opportunities for additional experiments to enhance the extraction of pavement characteristics through the OIM and eventually to transpose it for the processing of other kinds of signals, such as those from geophones.

## Figures and Tables

**Figure 1 sensors-22-05572-f001:**
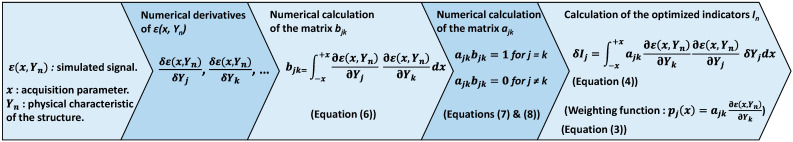
Flowchart for the determination of weighting functions and optimized indicators of a structure parameter.

**Figure 2 sensors-22-05572-f002:**
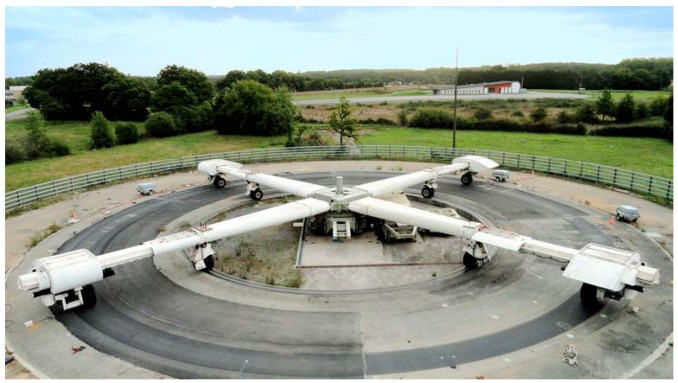
The pavement fatigue carousel at Univ. Eiffel Nantes [16].

**Figure 3 sensors-22-05572-f003:**
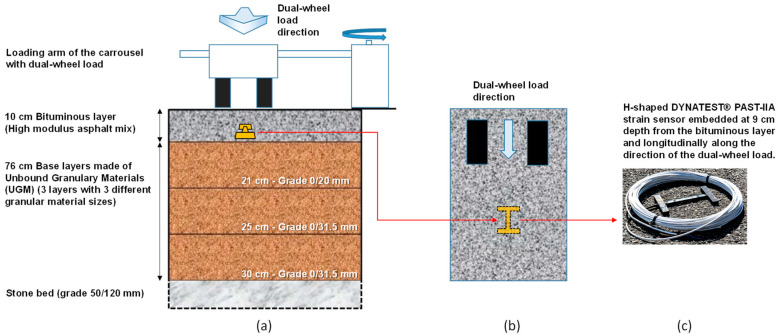
(**a**) Schematic representation in front view of the pavement’s structure, (**b**) top view illustrating the orientation of the strain gauge along the dual-wheel load direction, (**c**) picture of a DYNATEST^®^ strain gauge.

**Figure 4 sensors-22-05572-f004:**
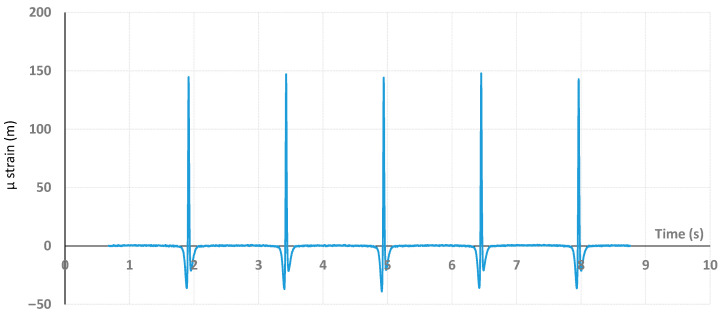
Example of µ strain signals recorded from the sensor as a function of time during spinning charging cycles.

**Figure 5 sensors-22-05572-f005:**
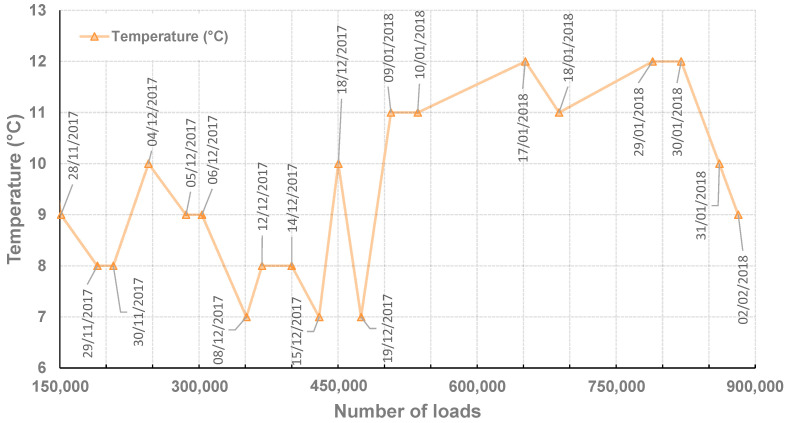
Mean temperatures and date of acquisition recorded inside the bituminous layer as a function of the number of loads.

**Figure 6 sensors-22-05572-f006:**
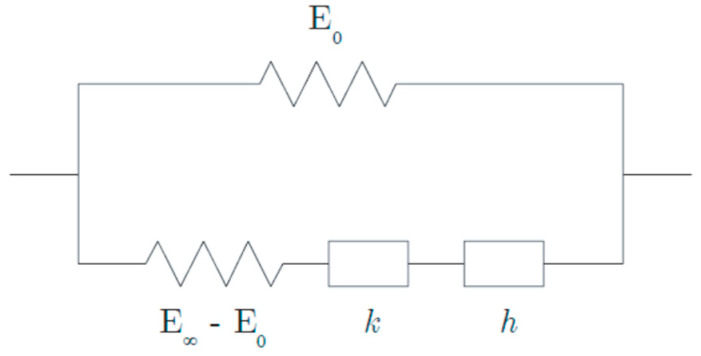
Huet–Sayegh model [22].

**Figure 7 sensors-22-05572-f007:**
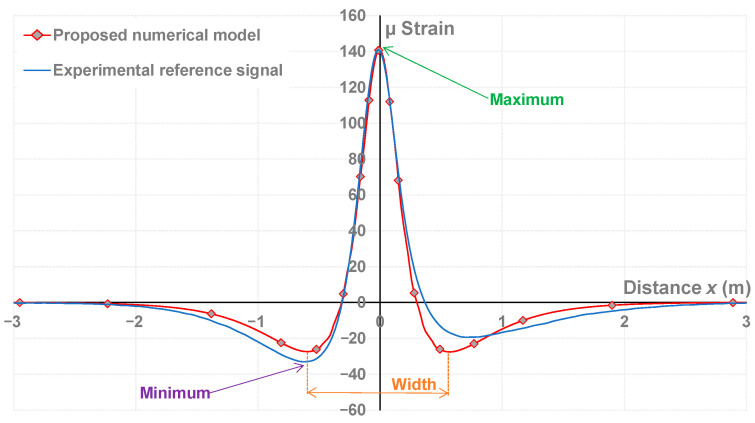
Fitting of the proposed numerical model to the reference signal.

**Figure 8 sensors-22-05572-f008:**
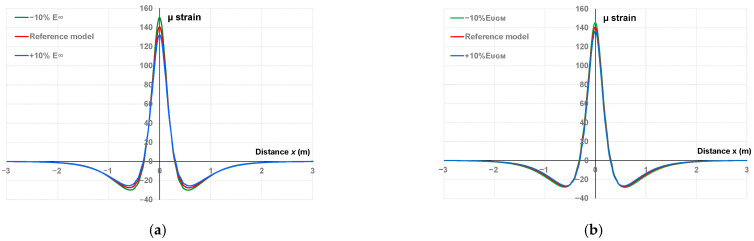
Fitting of the proposed numerical model to the reference signal with (**a**) ±10% variation for $E_{\infty }$, (**b**) ±10% variation for $E_{UGM}$.

**Figure 9 sensors-22-05572-f009:**
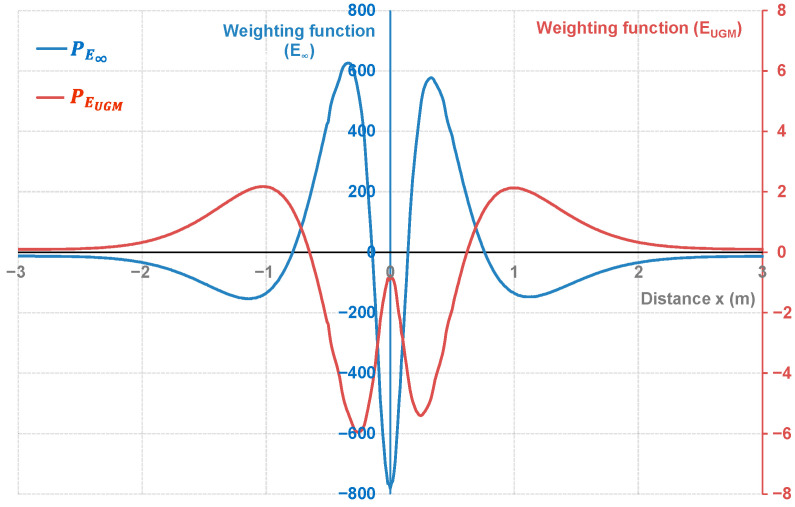
Weighting functions as a function of distance of $E_{\infty }$ (blue axis in the center) and $E_{UGM}$ (red axis in the right) obtained from the reference signal.

**Figure 10 sensors-22-05572-f010:**
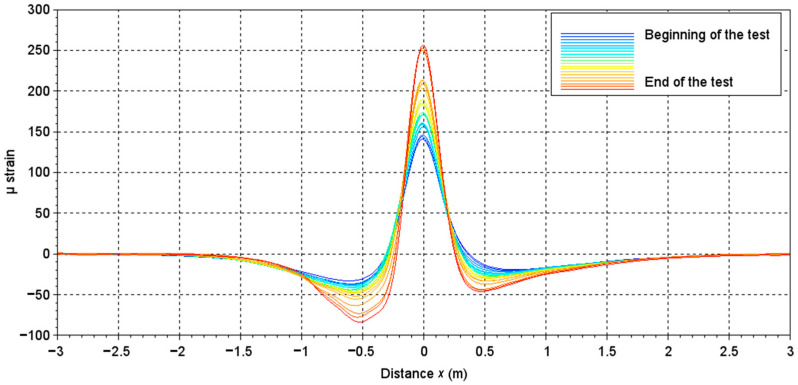
Evolution of gauge signals with the loading, from 151,200 loadings (beginning of the test) to 881,600 loadings (end of test).

**Figure 11 sensors-22-05572-f011:**
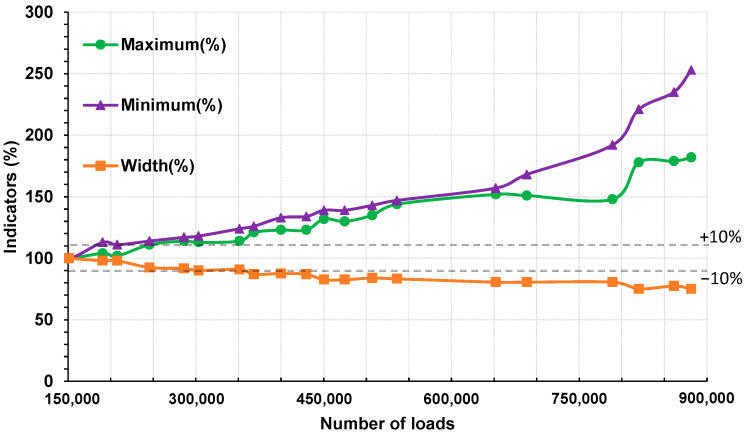
Relative evolution of the proposed indicators with the number of loads.

**Figure 12 sensors-22-05572-f012:**
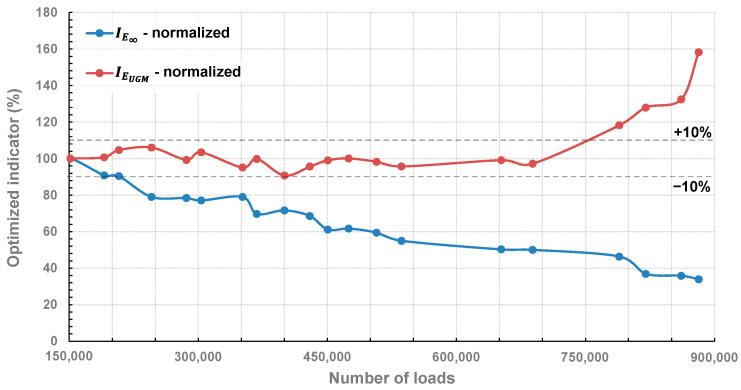
Relative evolution of the optimized indicators with the number of loads.

**Figure 13 sensors-22-05572-f013:**
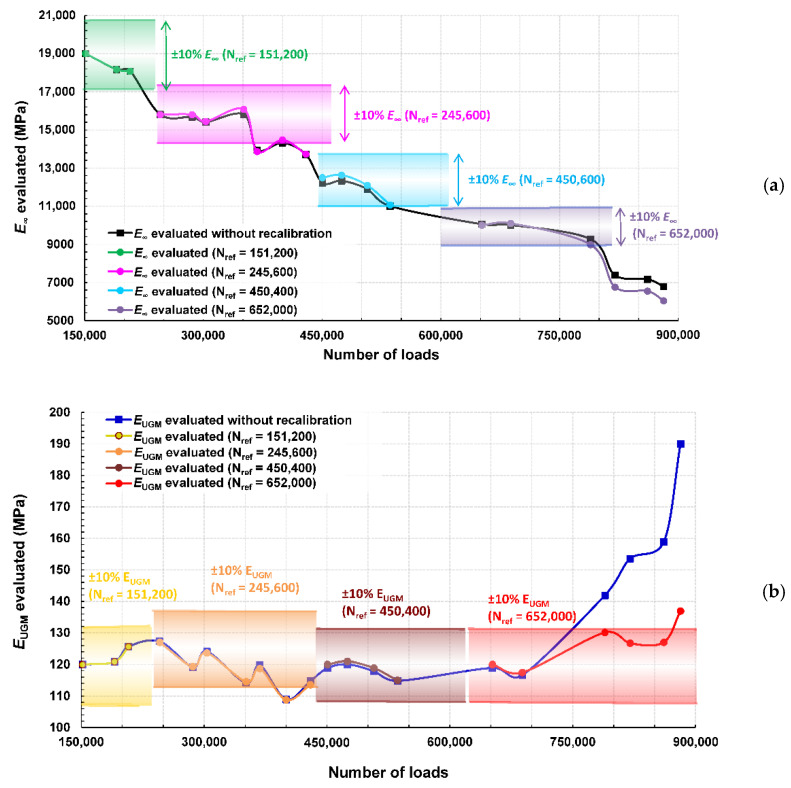
Evaluation of $E_{\infty }$ and $E_{UGM}$ as a function of loadings through the OIM with recalibration steps proceeding at numbers of loadings labeled as N_ref_ with ±10% variation windows (**a**) for $E_{\infty }$ and (**b**) for $E_{UGM}$.

**Table 1 sensors-22-05572-t001:** Pavement parameters for the proposed numerical model.

*E*_0_ (MPa)	$E_{\infty }$ (MPa)	$E_{UGM}$ (MPa)	*δ*	*h*	*k*	*τ*	*A* _0_	*A* _1_	*A* _2_
150	19,000	120	1.35	0.54	0.11	1.30	8.9	−0.44	0.0016

**Table 2 sensors-22-05572-t002:** Relative surface variation %S calculated after a ±10% variation for each parameter of the Huet–Sayegh model.

Parameter	*E* _0_	$E_{\infty }$	$E_{UGM}$	*δ*	*h*	*k*	*A* _0_	*A* _1_	*A* _2_
%S (%)	0	11	10	3	1	3	3	3	0

**Table 3 sensors-22-05572-t003:** Values and relative deviations between the different indicators.

	$E_{\infty }$		$E_{UGM}$	
	$I_{E_{\infty }}$	Deviation of the Indicator (%)	$I_{E_{UGM}}$	Deviation of the Indicator (%)
Reference signal	−18,098.81	0	−133.90	0
−10% $E_{\infty }$	−20,161.25	−11.46	−133.57	+0.25
+10% $E_{\infty }$	−16,361.25	+9.67	−133.57	+0.25
−10% $E_{UGM}$	−18,075.03	+0.13	−146.71	−9.34
+10% $E_{UGM}$	−18,075.03	+0.13	−122.71	+8.19

## Data Availability

Not applicable.

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
