# Peer review of "Definition of Optimized Indicators from Sensors Data for Damage Detection of Instrumented Roadways"

_sensors, 2022, doi:10.3390/s22155572_

Round 1
Reviewer 1 Report
The paper analyses the signals of strain gauges, conventionally and with the proposed OIM. It is a topic of interest to the researchers in the related areas but the paper needs some minor revision before acceptance for publication. My detailed comments are as follows:
1. Figure3(a) may be lack of grading curve, by the way, I think the configuration of temperature sensors should be better shown on this figure;
2. The title of Table1 is incorrect(Line231);
3. In section5.1, it can be seen from the Figure8(a)that both influence the maximum/minimum indicator, but it is not clear enough for me to recognize as one of the most important 2 factors. Some data especially the comparison between different indicators might be shown in this section instead of comparison with the indicator itself;
4. In Figure10, may miss on y-axis;
5. It’s interesting to investigate the local defect in both base layer and surface layer by OIM, but it would be better to provide the photo of defect situation in pavement (like asphalt core samples) after the experiment. It will help a lot for readers to understand the position and severity of the defect both in surface and base layer. It is also the practical evidence of this OIM method.
Author Response
Please see the attachement.

Reviewer 2 Report
Good information is presented. Please consider the following comments:
The tittle should be reformulated. “optimized indicators of ……?”
Extend your introduction.
Please add specific results in the abstract, highlighting the nobility of your research.
Line 34: Please add the complete forms when you mention an abbreviation, (Ground Penetrating Radar) GPR.
Line 46: It is not common to cite “To overcome this limitation, [6] have 46 developed an original”. Please mention his/her name.
What is the main methodological difference of this study and “Orthogonal Set of Indicators for the Assessment of Flexible Pavement Stiffness from Deflection Monitoring: Theoretical Formalism and Numerical Study”?
It is critical to clearly explain the main objectives of your research at the end of “introduction section”.
Line 65, 73, 182 and so on: error is citing style: “This method, developed and applied by ([6]) to obtain optimized”
Improve the quality and style of Figure 1. (Flowchart).
Line 112: it is not common to write a link in your text. Please use its correct citation form.
Add a reverence for figure 2. You take it from: https://lames.univ-gustave-eiffel.fr/en/equipments/the-pavement-fatigue-carrousel
what is “μstrain”? microstain?
Recheck the citation style of the journal.
Author Response
Please see the attachement

Round 2
Reviewer 2 Report
Most of the reforms were done. There is still problems with the citation style and abbreviations such as:
labeled “Optimized Indicators Method” (OIM),… Correct: Optimized Indicators Method (OIM)
M. Simonin & al. [12]
Please check and correct all according to the journal style.
Author Response
Please see the attachement.
